# The Comprehensive Characterization of B7-H3 Expression in the Tumor Microenvironment of Lung Squamous Cell Carcinoma: A Retrospective Study

**DOI:** 10.3390/cancers16112140

**Published:** 2024-06-04

**Authors:** Ayaka Asakawa, Ryoto Yoshimoto, Maki Kobayashi, Nanae Izumi, Takanori Maejima, Tsuneo Deguchi, Kazuishi Kubota, Hisashi Takahashi, Miyuki Yamada, Sachiko Ishibashi, Iichiroh Onishi, Yuko Kinowaki, Morito Kurata, Masashi Kobayashi, Hironori Ishibashi, Kenichi Okubo, Kenichi Ohashi, Masanobu Kitagawa, Kouhei Yamamoto

**Affiliations:** 1Department of Thoracic Surgery, Tokyo Medical and Dental University, 1-5-45 Yushima, Bunkyo-ku, Tokyo 113-8510, Japan; aasathsr@tmd.ac.jp (A.A.); masashikoba@gmail.com (M.K.); hishiba.thsr@tmd.ac.jp (H.I.); okubo.thsr@tmd.ac.jp (K.O.); 2Molecular Pathology Group, Translational Research Department, Daiichi Sankyo RD Novare, 1-16-13 Kitakasai, Edogawa-ku, Tokyo 134-8630, Japan; yoshimoto.ryoto.nc@rdn.daiichisankyo.co.jp (R.Y.); kobayashi.maki.c5@rdn.daiichisankyo.co.jp (M.K.); takahashi.hisashi.ff@rdn.daiichisankyo.co.jp (H.T.); yamada.miyuki.fm@rdn.daiichisankyo.co.jp (M.Y.); 3Translational Science Department, Daiichi Sankyo, Inc., Basking Ridge, NJ 07920, USA; nizumi@dsi.com (N.I.); kakubota@dsi.com (K.K.); 4Translational Science Department I, Daiichi Sankyo Co., Ltd., 1-2-58 Hiromachi, Shinagawa-ku, Tokyo 140-8710, Japan; maejima.takanori.yh@daiichisankyo.co.jp (T.M.); deguchi.tsuneo.e8@daiichisankyo.co.jp (T.D.); 5Department of Comprehensive Pathology, Graduate School of Medicine and Dentistry, Tokyo Medical and Dental University, 1-5-45 Yushima, Bunkyo-ku, Tokyo 113-8510, Japan; sishpth2@tmd.ac.jp (S.I.); iichpth2@tmd.ac.jp (I.O.); endpth2@tmd.ac.jp (Y.K.); kurata.pth2@tmd.ac.jp (M.K.); masa.pth2@tmd.ac.jp (M.K.); 6Department of Human Pathology, Graduate School of Medicine and Dentistry, Tokyo Medical and Dental University, 1-5-45 Yushima, Bunkyo-ku, Tokyo 113-8510, Japan; kohashi.pth1@tmd.ac.jp

**Keywords:** lung squamous cell cancer, B7-H3, PD-L1, multiplex immunohistochemistry, tumor microenvironment

## Abstract

**Simple Summary:**

Lung squamous cell carcinoma (LSCC) is refractory to various therapies for non-small cell cancer. The expression and significance of B7-H3 in the tumor microenvironment (TME) and its relationship with other immune checkpoint molecules have not yet been investigated. We used high-throughput quantitative multiplex immunohistochemistry to examine B7-H3 expression in the TME. We investigated the relationship between B7-H3 expression and prognosis as well as changes in the TME with B7-H3 expression. The correlation between B7-H3 and programmed cell death-ligand 1 (PD-L1) expression in single cells was also examined. A quantitative analysis of protein expression in macrophages and cancer cells revealed that PD-L1-positive cells expressed higher levels of B7-H3 than that of PD-L1-negative cells. Our findings demonstrate a correlation between B7-H3 and PD-L1 expression in the same cells, indicating that therapies targeting B7-H3 could provide additional efficacy in patients refractory to PD-L1-targeting therapies.

**Abstract:**

Lung squamous cell carcinoma (LSCC) is refractory to various therapies for non-small cell cancer; therefore, new therapeutic approaches are required to improve the prognosis of LSCC. Although immunotherapies targeting B7 family molecules were explored as treatments for several cancer types, the expression and significance of B7-H3 in the tumor microenvironment (TME) and its relationship with other immune checkpoint molecules have not yet been investigated in detail. We used high-throughput quantitative multiplex immunohistochemistry to examine B7-H3 expression in the TME. We investigated the relationship between B7-H3 expression and prognosis as well as changes in the TME with B7-H3 expression using 110 surgically resected pathological specimens retrospectively. We examined the correlation between B7-H3 and programmed cell death-ligand 1 (PD-L1) expression in single cells. High B7-H3 expression in tumor cells was associated with a better prognosis and a significant increase in the number of CD163^+^PD-L1^+^ macrophages. Quantitative analysis revealed that there is a positive correlation between B7-H3 and PD-L1 expression in tumor and stromal cells, as well as in intratumoral tumor-infiltrating lymphocytes and tumor-associated macrophages in the same cells. CD68^+^, CD163^+^, and CK^+^ cells with PD-L1^+^ phenotypes had higher B7-H3 expression compared to PD-L1^−^ cells. Our findings demonstrate a correlation between B7-H3 and PD-L1 expression in the same cells, indicating that therapies targeting B7-H3 could provide additional efficacy in patients refractory to PD-L1-targeting therapies.

## 1. Introduction

Lung cancer is one of the most common cancers globally and is the main cause of cancer-related deaths [1]. Although treatments for non-squamous cell lung cancer have improved considerably in recent decades, no improvement in therapies for lung squamous cell carcinoma (LSCC) has been reported. Immune checkpoint inhibitor (ICI) therapies were developed to treat various cancers [2,3]. Particularly, immunotherapies targeting B7 family molecules such as programmed cell death-ligand 1 (PD-L1) were explored as treatments for several cancer types [4,5], including non-small cell lung cancer (NSCLC). Consequently, immunotherapy is now a major treatment modality for patients with NSCLC [6].

The B7 family comprises 10 members: CD80; CD86; B7-H1 (PD-L1); B7-DC (PD-L2); B7-H2 (ICOSL); B7-H3 (CD 276); B7-H4; B7-H5; B7-H6; and B7-H7. These molecules provide positive and negative signals that stimulate and suppress T-cell activity, respectively [5]. B7-H3 is a type 1 transmembrane protein that acts as an immunoregulatory molecule and has attracted attention as a target for ICIs [7,8]. Although the B7-H3 protein is known to be expressed in activated dendritic cells, monocytes, T cells, B cells, natural killer (NK) cells, tumor-infiltrating lymphocytes (TILs), and tumor-associated macrophages (TAMs), which affect tumor growth, metastasis, and drug resistance [5,9], its immunological functions remain unclear as it affects T-cell activation and inhibition as well as NK cell responses. Furthermore, the effects of B7-H3 localization are unknown as B7-H3 is expressed in both tumor cells and peritumoral stromal cells [10].

Therapies targeting B7-H3 have been hypothesized to exert synergistic effects with current PD-L1 inhibitors or be used as an additional ICI therapy for cancers that are refractory to PD-L1 inhibition or express low levels of PD-L1. However, the relationship between prognosis and B7-H3 expression in patients with LSCC under standard treatment remains unclear. Furthermore, the quantitative dynamics and crosstalk between the tumor microenvironment (TME) and stromal areas in LSCC are unknown, and no studies to date have examined the relationship between B7-H3 and the expression of other ICI targets, such as PD-L1, at the cellular level. Therefore, we aimed to determine the clinicopathological significance of B7-H3 expression by examining the relationship between B7-H3 expression in cancer cells and the TME and prognosis in patients with LSCC under existing treatments. Additionally, we sought to explore the quantitative relationship between B7-H3 expression in the tumor, stroma, and TME cells using advanced multiplex immunohistochemical staining and analytical techniques. Finally, we aimed to determine whether ICIs targeting B7-H3 could improve the efficacy of existing ICIs by examining the relationship between the expression of other ICI target molecules and B7-H3 in cell subsets.

## 2. Materials and Methods

### 2.1. Patients and Methods

Patients who underwent radical lung resection for stage II or III LSCC at the Tokyo Medical and Dental University between August 2010 and March 2019 were included in this study. A total of 111 patients were a part of this study. The sex was adopted as the sex listed on the health insurance card administered by Japanese government. The number of cases was determined based on the hypothesis that significant differences would occur when patients were divided into two groups. The clinicopathological characteristics of the cohort were collected retrospectively based on patient medical records and surgical pathology reports. Patients who underwent reoperation, preoperative chemotherapy, or radiotherapy were excluded. No exclusions were present that would conflict with the randomization other than the exclusion criteria. Pathological diagnosis was confirmed according to the 8th Edition of the TNM Classification staging system [11]. Informed consent was obtained from all patients in the form of documents or opt-outs. Ethical approval was obtained from the Institutional Review Board of Tokyo Medical and Dental University (approval number: M2019-177), Daiichi Sankyo Co., Ltd. and Daiichi Sankyo RD Novare Co., Ltd. (1-16-13 Kitakasai, Edogawa-ku, Tokyo, Japan; approval number: J-20-015(00417-06)).

### 2.2. Immunohistochemistry

Formalin-fixed paraffin-embedded (FFPE) tissue blocks were cut into 4 µm thick sections. For single immunohistochemistry (sIHC) experiments, anti-B7-H3 (BD5A11-ocChimera, Daiichi Sankyo, Tokyo, Japan) primary antibodies were used. Sections were stained with VENTANA BenchMark ULTRA using a VENTANA OptiView DAB IHC detection kit (Roche Diagnostics, Tucson, AZ, USA).

For multiplex immunohistochemistry (mIHC), FFPE sections (4 µm thick) were stained using the Opal^TM^ Polaris 7-Color Automation IHC Kit (Akoya Biosciences, Marlborough, MA, USA) with primary antibodies against B7-H3 (mouse monoclonal antibody, 1:600, BD5A11, Daiichi Sankyo), PD-L1 (1:100, SP142, abcam, Cambridge, MA, USA), PD-1 (1:200, NAT105, abcam), CD4 (ready-to-use [RTU], 4B12, Leica Biosystems, Nußloch, Germany), CD8 (RTU, 4B11, Leica Biosystems), CD68 (RTU, 514H12, Leica Biosystems), CD163 (RTU, 10D6, Leica Biosystems), HLA class I (1:800, EMR8-5, abcam), and pan-cytokeratin (RTU, AE1/AE3, Leica Biosystems), according to the manufacturer’s instructions. All antibodies were stained using sequential protocols. All sections were stained using BOND RX (Leica Biosystems) (Figure 1).

### 2.3. Evaluation of B7-H3 Intensity

B7-H3 expression was evaluated using sIHC slides. The staining intensity of membrane B7-H3 expression was assessed in tumor and stromal cells, and patients were divided into two groups accordingly. Tumor cell B7-H3 staining was assessed using the HER2 IHC scoring method, as described earlier with minor modifications [12]. The staining strength was assessed by two pathologists, who then discussed their assessment and made a staining decision. If tumor cells with evident membrane B7-H3 staining accounted for >30% of the tumor area, the patient was assigned to the high-expression group, whereas all other patients were assigned to the low-expression group (Figure 2a,b). Stromal cell staining intensity was defined as either strong or weak. If more cells had strong staining intensity, the patient was assigned to the high-expression group, whereas all other patients were assigned to the low-expression group (Figure 2c,d).

### 2.4. Analysis of mIHC Slides

All slides were subjected to seven-color multispectral image analysis using an Automated Quantitative Pathology Imaging System (Vectra Polaris; Akoya Biosciences, Marlborough, MA, USA). Whole-slide images were scanned at 10× magnification. For quantitative imaging, the top five TIL hotspot fields were selected as regions of interest (ROIs) using Phenochart Viewer (version 1.0.12, Akoya Biosciences). The same ROIs were selected for TAM analysis. After scanning, slide data were analyzed using inForm Automated Image Analysis Software (version 2.5.1. Akoya Biosciences) with the Trainable Tissue Segmentation, Adaptive Cell Segmentation, and Phenotyping functions. After phenotypes were assigned to individual cells, batch analyses were performed using inForm software. Data were consolidated and analyzed using Phenoptr (version 0.2.9) and PhenoptrReports (version 0.2.10, Akoya Biosciences) software. Cells in stromal and intratumoral regions were classified into the following phenotypes: TILs (CD4^+^/PD-1^−^, CD4^+^/PD-1^+^, CD8^+^/PD-1^−^, and CD8^+^/PD-1^+^ cells); TAMs (CD68^+^/PD-L1^−^, CD68^+^/PD-L1^+^, CD163^+^/PD-L1^−^, and CD163^+^/PD-L1^+^ cells); and tumor cells (B7-H3^−^/PD-L1^−^/CK^+^, B7-H3^−^/PD-L1^+^/CK^+^, B7-H3^+^/PD-L1^−^/CK^+^, and B7-H3^+^/PD-L1^+^/CK^+^ cells).

### 2.5. Evaluation of Membrane B7-H3 Expression

Cell membrane B7-H3 expression was estimated for each pixel using inForm, Phenoptr, and PhenoptReports software, as described above. B7-H3 expression levels were compared based on the different cell phenotypes.

### 2.6. Statistical Analysis

Kaplan–Meier survival curves were used to compare the overall survival between patients in the high- and low-B7-H3 expression groups. Fisher’s exact test was used to compare clinicopathological features between the groups. Univariate and multivariate analyses were performed using the Cox proportional hazard regression test. The correlation between membrane B7-H3 and PD-L1 expression or intensity was calculated using Spearman’s correlation analysis. Mann–Whitney *U*-tests were used to determine the significance of differences in B7-H3 expression for each phenotype.

GraphPad Prism (version 9.1.0, GraphPad Software, San Diego, CA, USA) was used to perform Kaplan–Meier, correlation, and Mann–Whitney *U* analyses. Log-rank tests and Cox regression analyses were performed using EZR version 1.50 (Saitama Medical Center, Jichi Medical University, Saitama, Japan), a graphical user interface for R (The Foundation for Statistical Computing, Vienna, Austria) [13]. All differences were considered statistically significant at *p* < 0.05.

## 3. Results

### 3.1. Patient Characteristics

Lung cancer specimens were collected from 111 patients who underwent radical lung resection for stage II or III LSCC at the Tokyo Medical and Dental University. The median age was 70 years (range: 45–82 years), and the cohort included 84 male and 27 female patients. Based on sIHC and mIHC staining for B7-H3, patients with widely varying B7-H3 staining properties were excluded (*n* = 8) because of the poor reproducibility of staining intensity. The remaining 103 patients had a median follow-up time of 25.6 months (range: 0.7–104.3 months). sIHC analysis showed that 46 patients (44.7%) had high tumor B7-H3 expression, and 57 patients (55.3%) of 103 patients had low tumor B7-H3 expression. Furthermore, 48 (46.6%) and 55 (53.4%) patients of 103 patients had high and low stromal B7-H3 expressions, respectively. The patient characteristics of each group are shown in Table 1. The Brinkman index, which is calculated by multiplying the number of cigarettes smoked per day by the number of years the person has been smoking, was significantly higher in the high-tumor B7-H3 expression (*p* = 0.0258) and high-stromal B7-H3 expression (*p* = 0.0142) groups than that in the low-expression group. Other parameters did not differ between the groups.

### 3.2. Prognostic Significance of B7-H3 Expression

To investigate the relationship between B7-H3 expression and lung cancer prognosis, Kaplan–Meier survival analyses were performed. In the analyses, patients with high tumor B7-H3 expression had a significantly better prognosis than those with low tumor B7-H3 expression (*p* = 0.0341); however, stromal B7-H3 expression was not statistically associated with 5-year overall survival (Figure 2e–g).

### 3.3. Relationship between Clinicopathological Features and Survival

To investigate the relationship between the clinicopathological features and 5-year survival of the patients, univariate and multivariate analyses were performed. Univariate analysis revealed that tumor B7-H3 expression (*p* = 0.034) and lymphovascular invasion (*p* = 0.04) were associated with 5-year survival. Multivariate analysis revealed that only tumor B7-H3 expression (*p* = 0.045) was significantly associated with survival, Table 2.

### 3.4. Relationship between Tumor B7-H3 Expression and Cell Phenotypes

To determine whether tumor B7-H3 expression associated the pattern of cell phenotypes, we compared the cell density of each cell phenotype in samples from patients with high and low tumor B7-H3 expression. Among the cell density of TILs (CD4^+^PD-1^−^, CD4^+^PD-1^+^, CD8^+^PD-1^−^, and CD8^+^PD-1^+^ cells) and TAMs (CD68^+^PD-L1^−^, CD68^+^PD-L1^+^, CD163^+^PD-L1^−^, and CD163^+^PD-L1^+^ cells), only CD163^+^PD-L1^+^ cells in the stromal area were significantly higher in number in the high-expression group (*p* = 0.0216), whereas other phenotypes were not significantly different between the groups (Figure 3).

### 3.5. Relationship between Stromal B7-H3 Expression and Cell Phenotypes

Next, we determined whether stromal B7-H3 expression was also associated with different cell phenotypes by measuring the cell densities of TILs (CD4^+^PD-1^−^, CD4^+^PD-1^+^, CD8^+^PD-1^−^, and CD8^+^PD-1^+^ cells) and TAMs (CD68^+^PD-L1^−^, CD68^+^PD-L1^+^, CD163^+^PD-L1^−^, and CD163^+^PD-L1^+^ cells). The counts of CD8^+^PD-1^−^ cells were significantly higher in the tumor area in the low-B7-H3 expression group than those in the high-B7-H3 expression group (*p* = 0.0262); however, no other phenotypes were significantly associated in either area (Appendix A).

### 3.6. Correlation between B7-H3 and PD-L1 Expression

Finally, we examined the correlation between membrane B7-H3 and PD-L1 expression in tumor and stromal cells by examining B7-H3 expression or PD-L1 expression levels per cell membrane pixel with slides from all groups (Figure 4). B7-H3^+^ expression correlated positively with PD-L1 expression in tumor and other cells in the TME, including intratumoral TILs and TAMs (Figure 4a,b). Additionally, we observed a positive correlation between membrane B7-H3 and PD-L1 expression in stromal cells (Figure 4c).

To determine the phenotypes of cells expressing B7-H3, the membrane B7-H3 expression levels per cell membrane pixel were calculated for each phenotype. CD68^+^, CD163^+^, and CK^+^ cells with PD-L1^+^ phenotypes had higher B7-H3 expression than that of PD-L1^-^ cells, although PD-1^+^ phenotypes were not associated with B7-H3 expression (Figure 5). Together, these results indicate a positive correlation between PD-L1 expression and upregulated B7-H3 expression.

## 4. Discussion

LSCC is refractory to many standard therapies for NSCLC; therefore, new therapeutic approaches are required to improve its prognosis. Although immunotherapies targeting B7 family molecules have been explored for treating several types of cancer, the expression and significance of B7-H3 in the TME and its relationship with other immune checkpoint molecules have not yet been investigated in detail. Here, we compared the prognosis of LSCC among patients grouped according to high or low B7-H3 expression in tumor or stromal cells. High tumor B7-H3 expression was associated with a favorable prognosis of LSCC, whereas stromal B7-H3 expression was not associated with a prognosis at all. High tumor B7-H3 expression is linked with a poorer prognosis in patients with NSCLC [14,15,16], contrary to the findings of our study. This finding could be attributed to the fact that previous reports examining B7-H3 expression in NSCLC have mainly focused on adenocarcinomas [17], whereas we examined LSCC. Adenocarcinoma and squamous cell carcinoma differ in terms of both patient backgrounds, including sex and smoking history, and in indications for tyrosine kinase inhibitors [18], which may explain the observed differences in prognosis. The present cohort includes many smokers. Smoking-related cancers may express larger amounts of neoantigen [19]. If the analysis were performed in a cohort of patients with ICIs, different results could be obtained. It is also possible that the roles of B7-H3 in recurrence differ between adenocarcinomas and squamous cell carcinomas. For instance, it has been suggested that the overexpression of B7-H3 was associated with lymph node and distant metastasis in lung adenocarcinoma [20]. High B7-H3 expression has been reported as a favorable prognostic factor in patients with squamous cell carcinoma of the head and neck [21]. Furthermore, there are two isoforms of B7-H3 [22], but it has not been confirmed which isoform the B7-H3 used in this study recognizes. If this antibody were to recognize isoform 2, it would be expected to attack tumor cells by enhancing CD8-positive T cells, resulting in a better prognosis for the patient, which is consistent with the results of this study. This may be one of the reasons why the results of this study differed from those previously reported. Thus, this controversial role of B7-H3 in the progression and recurrence of squamous cell carcinoma may also explain the difference in prognosis between LSCC and adenocarcinoma.

In this study, we expected that relatively low stromal B7-H3 levels would result in a more active immune response to the tumor and a stronger tumor-killing effect associated with a better prognosis; however, we found that stromal B7-H3 expression was not associated with prognosis. This may be related to the interaction of cells in the TME, including TILs and TAMs.

When we stratified tumor and stromal TILs and TAMs by B7-H3 expression, we found no significant difference in intratumoral CD8^+^ T-cell infiltration in the group with low tumor B7-H3 expression, whereas the group with low stromal B7-H3 expression had a significantly higher density of intratumoral CD8^+^ T cells. B7-H3 inhibits T-cell proliferation [23,24] and is associated with the inhibition of CD8^+^ T-cell upregulation, whereas aberrant B7-H3 expression is implicated in the elimination and dysfunction of CD8^+^ T-cell infiltration in head and neck cancer [25]. These findings are consistent with the results of this study, wherein we observed an increase in the density of CD8^+^ TILs in tumors with low B7-H3 expression. In NSCLC, tumor cells expressing B7-H3 may promote immune tolerance by avoiding CD8^+^ T-cell damage [21,26]. Therefore, low stromal B7-H3 levels in LSCC may activate the local immune system to promote CD8^+^ T-cell invasion.

We also found that CD163^+^PD-L1^+^ macrophages were significantly more prevalent in tumors with high B7-H3 expression, which was associated with a good prognosis. Generally, patients with more TAM infiltration around tumor cells have a poorer prognosis, and CD163^+^ macrophage infiltration is associated with a poor prognosis in squamous cell carcinoma [27,28]. CD163^+^ macrophages affect PD-L1 expression in tumor cells, and CD163^+^ M2 macrophages exert immunosuppressive effects by activating JAK/STAT signaling to produce IL-10 and promote PD-L1 expression in tumor cells [28]. Various mechanisms were hypothesized for the emergence of macrophages that co-express CD163 and PD-L1, including the proposed mechanism wherein B7-H3 expression by tumor cells reduces the peritumoral immune response and induces CD163^+^ macrophages. CD163^+^ macrophages are speculated to receive IL-10 signals produced by their own cells or similar macrophages in the surrounding area and subsequently to co-express PD-L1 via the JAK/STAT pathway. However, the mechanism of a subset of specialized macrophages expressing CD163^+^PD-L1^+^ that would act against tumors expressing high B7-H3 levels remains unclear. As this was not observed in patients with high stromal or tumor B7-H3 expression, it is possible that specific signals and interactions are required for this phenomenon to occur, such as maintaining an appropriate distance between B7-H3 signals and macrophages in the tumor.

B7-H3 and PD-L1 are both present on the plasma membranes of tumor and antigen-presenting cells [5], and their co-expression was previously noted in other cancers [29,30]. In primary lung cancer, B7-H3 tends to be co-expressed with PD-L1 in LSCC cells [30]. However, as these studies evaluated B7-H3 and PD-L1 expression in the entire tumor, the relationship between B7-H3 and PD-L1 protein expression at the single-cell level was unknown. Here, we observed quantitative correlations between B7-H3 and PD-L1 expression on the plasma membrane of stromal cells, including tumor, and immune cells. Thus, the objective of this study is the first to confirm this phenomenon at the cellular level. Multiplex IHC enabled the quantitative evaluation of expression levels in the same cell, and we revealed that the presence or absence of PD-L1 expression correlates with B7-H3 expression in stromal cells, including TAMs, at the single-cell level. These results suggest that both tumor and stromal cells may upregulate B7-H3 and PD-L1 expression via the same mechanism or that the expression of one factor upregulates the expression of the other. In any case, the observed similarity in the expression of proteins with immunological functions in both tumor and stromal cells holds great promise for therapeutic applications, and the mechanism of co-expression warrants further investigation.

In patients with NSCLC, the current outcome of PD-L1 treatment is 84.3% when treatment can be completed, but the outcome is only 27.3–39.1% in cases where treatment cannot be completed [31]. This indicates that the prognosis is good if treatment is completed, while the prognosis is poor if treatment is not completed. Therefore, the co-expression of B7-H3 and PD-L1 in tumor and stromal cells could provide various clinical benefits. For instance, B7-H3 ICI therapy could be used to treat PD-L1-expressing tumors that do not respond adequately to PD-L1 ICI therapy, as combination therapy with two ICIs has already been used to treat NSCLC with other B7 family ligands. Alternatively, B7-H3 ICIs could be used to treat tumors resistant to existing ICIs [32,33]. The efficacy of treatments can be improved by combining two types of ICIs from the beginning [34,35]. Therefore, targeting B7-H3 and PD-L1 using ICIs could be a promising therapeutic strategy. These are merely hypotheses, and further research is needed to substantiate them.

Despite these promising findings, this study has several limitations. The study cohort included patients who received multiple postoperative treatments, including six ICI cases, and it is possible that the different response rates may have affected prognosis in both patient groups. Therefore, a large prospective cohort survey is required to verify our findings. We were also unable to examine changes in the TME before and after ICI treatment in this study, and although we demonstrated a correlation between PD-L1 and B7-H3 expression, the molecular mechanism involved in correlation remains unclear and should be clarified in the future. The proportion of tumor and stromal regions in ROIs is not uniform. A rigorous adjustment of the proportions could provide more reliable data. Furthermore, several markers were examined in the quantitative search for stromal cells; however, it was not possible to stain some markers for stromal cells, including vascular endothelial cells, fibroblasts, and blood cells, which may also have been affected by B7-H3 expression.

## 5. Conclusions

In summary, this study demonstrates the relationship between B7-H3 expression and the TME as well as the quantitative correlation between PD-L1 and B7-H3 expression at the single-cell level and the clinicopathological effects of B7-H3 protein expression in LSCC. Our findings improve our understanding of the complex relationship between LSCC and the TME and suggest that B7-H3 ICI therapies could improve the prognosis of lung cancer following further research and development.

## Figures and Tables

**Figure 1 cancers-16-02140-f001:**
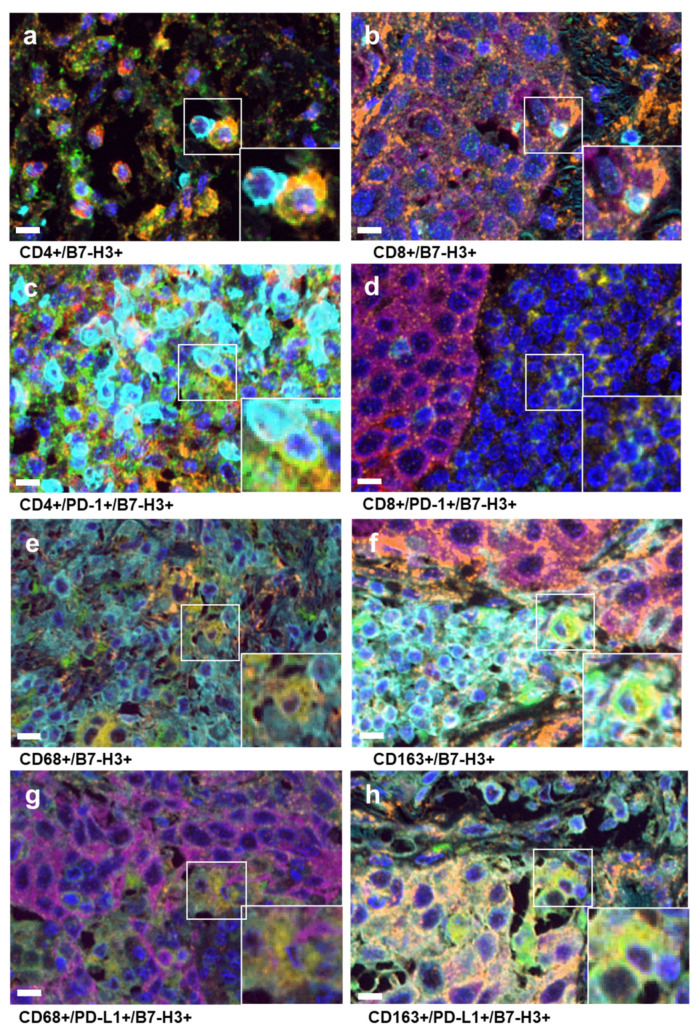
Results of multiplex immunohistochemistry. (**a**–**d**) B7-H3: orange, PD-L1: red, CD4: yellow–green, CD8: light blue, PD-1: yellow, CK: pink. (**e**–**h**) B7-H3: orange, PD-L1: red, CD163: yellow–green, HLA: light blue, CD68: yellow. (Asakawa et al.).

**Figure 2 cancers-16-02140-f002:**
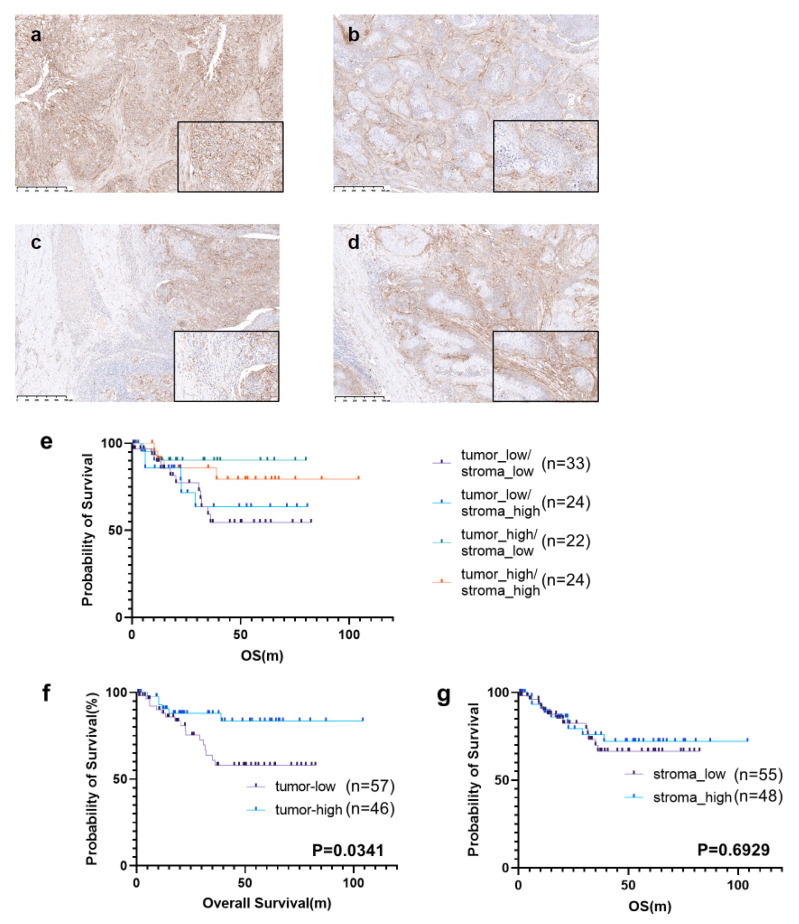
Single staining of B7-H3 and Kaplan–Meier curve stratified by B7-H3 expression. (**a**–**d**) Single staining of B7-H3. (**a**) High tumor B7-H3 expression. (**b**) Low tumor B7-H3 expression. (**c**) Low stromal B7-H3 expression. (**d**) High stromal B7-H3 expression. (**e**–**g**) Kaplan–Meier curve based on tumor and/or stromal B7-H3 expression. (Asakawa et al.).

**Figure 3 cancers-16-02140-f003:**
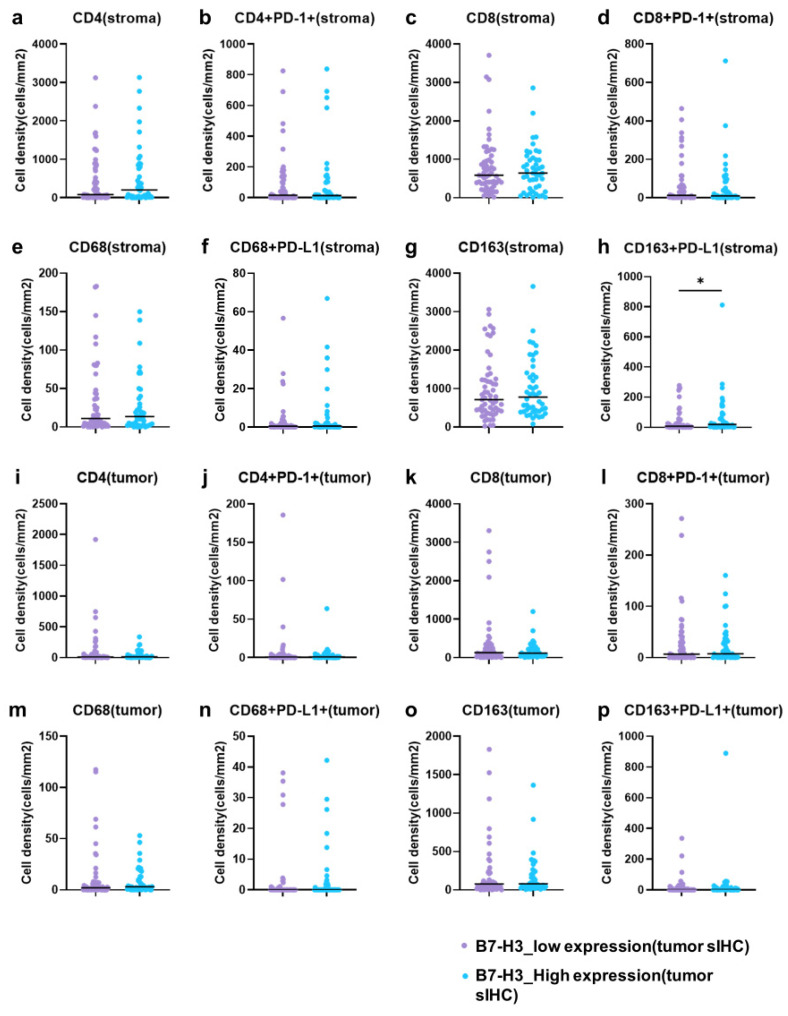
Relationship between tumor B7-H3 expression and cell phenotypes. (**a**) Stromal CD4^+^ single-positive cells. (**b**) Stromal CD4^+^PD-1^+^ double-positive cells. (**c**) Stromal CD8^+^ single-positive cells. (**d**) Stromal CD8^+^PD-1^+^ double-positive cells. (**e**) Stromal CD68^+^ single-positive cells. (**f**) Stromal CD68^+^PD-L1^+^ double-positive cells. (**g**) Stromal CD163^+^ single-positive cells. (**h**) Stromal CD163^+^PD-L1^+^ double-positive cells. (**i**) Tumor CD4^+^ single-positive cells. (**j**) Tumor CD4^+^PD-1^+^ double-positive cells. (**k**) Tumor CD8^+^ single-positive cells. (**l**) Tumor CD8^+^PD-1^+^ double-positive cells. (**m**) Tumor CD68^+^ single-positive cells. (**n**) Tumor CD68^+^PD-L1^+^ double-positive cells. (**o**) Tumor CD163^+^ single-positive cells. (**p**) Tumor CD163^+^PD-L1^+^ double-positive cells. The horizontal bars indicate the median, and the vertical axis of the graph represents the cell density. (Asakawa et al.). * *p* < 0.1.

**Figure 4 cancers-16-02140-f004:**
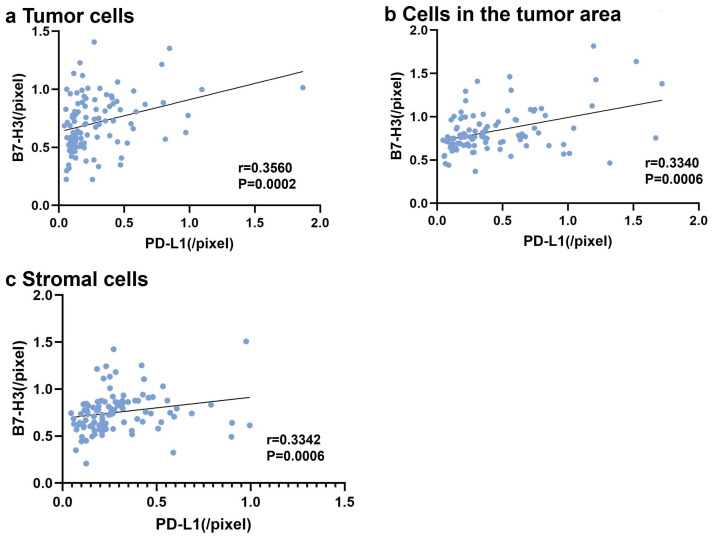
Correlations between membrane B7-H3 and PD-L1 expression. The horizontal axis represents PD-L1 expression levels per cell membrane pixel, and the vertical axis represents B7-H3 expression levels per cell membrane pixel. (**a**) Tumor cells. (**b**) Cells in the tumor area other than cancer cells, including intratumoral tumor-infiltrating lymphocytes (TILs) and tumor-associated macrophages (TAMs). (**c**) Stromal cells. (Asakawa et al.).

**Figure 5 cancers-16-02140-f005:**
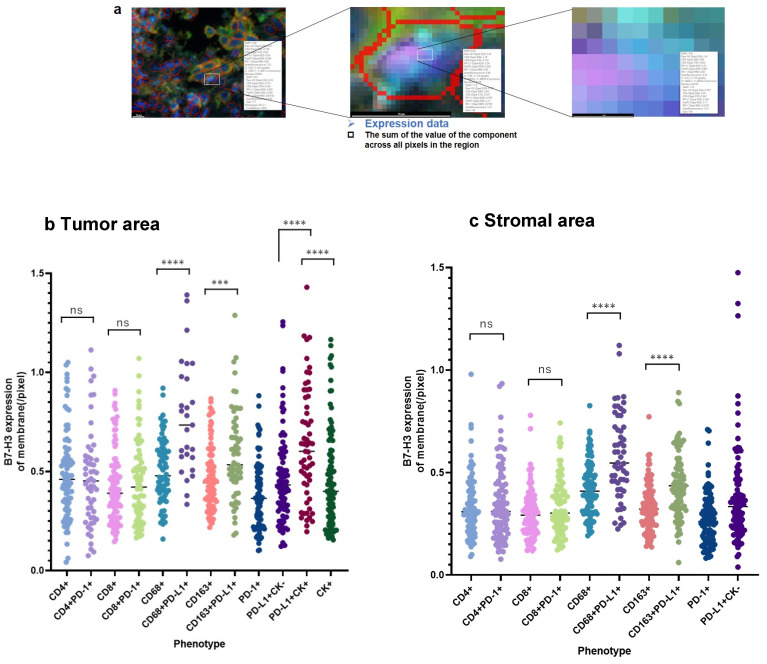
Expression of B7-H3 at the cell membrane. (**a**) A scheme for calculating B7-H3 expression levels per cell membrane pixel. (**b**) Membrane B7-H3 expression for each phenotype cell in the tumor area. The vertical axis presents B7-H3 expression per cell membrane pixel. (**c**) The B7-H3 expression of the cell membrane for each phenotype cell in the stromal area. The vertical axis presents B7-H3 expression per cell membrane pixel. *** *p* < 0.001; **** *p* < 0.0001; ns: not significant.

**Table 1 cancers-16-02140-t001:** The clinicopathological background of the patients stratified by tumor and stromal B7-H3 expression (*n* = 103).

		Tumor B7-H3			Stromal B7-H3		
Characteristics	*n* (%)	High (*n* = 46)	Low (*n* = 57)	*p* Value	High (*n* = 48)	Low (*n* = 55)	*p* Value
Age (median)				0.1667			>0.9999
>70	51 (49.5)	19	32		24	27	
≤70	52 (50.5)	27	25		24	28	
Gender				0.115			0.371
Male	77 (74.8)	38	39		38	39	
Female	26 (25.2)	8	18		10	16	
Stage				0.4136			0.8368
II	67 (65.0)	32	35		32	35	
III	36 (35.0)	14	22		16	20	
Brinkman Index				0.0258			0.0142
>600	76 (73.8)	39	37		41	35	
≤600	27 (26.2)	7	20		7	20	
ly				>0.9999			0.8207
+	25 (24.3)	11	14		11	14	
−	78 (75.7)	35	43		37	41	
v				0.6328			0.1503
+	81 (78.6)	35	46		41	40	
−	22 (21.4)	11	11		7	15	
pl				0.4251			0.6941
+	60 (58.3)	29	31		29	31	
−	43 (41.7)	17	26		19	24	
pm				0.7289			0.7209
+	8 (7.8)	3	5		3	5	
−	95 (92.2)	43	52		45	50	

ly: lymphovascular invasion, v: vascular invasion, pl: pleural invasion, pm; intrapulmonary metastasis.

**Table 2 cancers-16-02140-t002:** Univariate and multivariate analyses of clinicopathological factors related to overall survival. ly; lymphovascular invasion, v: vascular invasion, pl; pleural invasion, pm; intrapulmonary metastasis.

Univariate Analysis				Multivariate Analysis		
		*n*	*p* Value	Hazard Ratio	95% CI	*p* Value
B7-H3 (tumor)	High	46	0.034	0.3855	0.1519, 0.9786	0.04492
	Low	57				
B7-H3 (stroma)	High	48	0.693			
	Low	55				
Gender	Male	77	0.705			
	Female	26				
Age	>70	51	0.299			
	≤70	52				
Stage	II	67	0.449			
	III	36				
Brinkman Index	>600	76	0.858			
	≤600	27				
ly	+	25	0.04	0.9954	0.9954, 5.1900	0.05129
	−	78				
v	+	81	0.284			
	−	22				
pl	+	60	0.572			
	−	43				
pm	+	8	0.062			
	−	95				

## Data Availability

Raw data for this study were generated by the Daiichi Sankyo RD Novare Company (Tokyo, Japan). The data supporting the findings of this study are available from the corresponding author upon request.

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
