# Peer review of "The Comprehensive Characterization of B7-H3 Expression in the Tumor Microenvironment of Lung Squamous Cell Carcinoma: A Retrospective Study"

_cancers, 2024, doi:10.3390/cancers16112140_

Round 1

Reviewer 1 Report

Comments and Suggestions for Authors

This manuscript investigated the B7-H3 expression in tumor microenvironment and explored the potential correlation between B7-H3 and PD-L1 in lung squamous cell carcinoma. This study is well-designed t reveal the B7-H3 expression at the single-cell level. The manuscript is interesting to publication in 'cancers' after revision.

1) It could be better to show the quantitative results of single staining of B7-H3 expression using QuPath or other software for Figure 2.

2) In section 3.1, It will be better to show the quantitative mIHC and sIHC results instead of only present the percentage.

3) In table 1, the subtitle and its content is not well designed. For example, the subtitle "P-vale" is located in the middle of two numbers.

4) Authors show the correlation between B7-H3 and PD-L1 expression. Whether the data is from the whole group or its from high B7-H3 expression group?

Both B7-H3 and PD-L1 belongs to B7 family, it will not be very surprised to observe the positive correlation expression of B7-H3 and PD-L1. In addition, it looks like no significant difference between tumor, TMC, and stromal cells in Figure 4. Could authors claim the novelty or significance of this results?

Comments on the Quality of English Language

Minor editing of English language required

Author Response

Reviewer 1

This manuscript investigated the B7-H3 expression in tumor microenvironment and explored the potential correlation between B7-H3 and PD-L1 in lung squamous cell carcinoma. This study is well-designed t reveal the B7-H3 expression at the single-cell level. The manuscript is interesting to publication in 'cancers' after revision.

1) It could be better to show the quantitative results of single staining of B7-H3 expression using QuPath or other software for Figure 2. 

>Thank you for the very useful suggestion. Manual evaluation was chosen in our laboratory due to a lack of technical equipment and expertise in dealing with the QuPATH software. To compensate for this shortcoming, two pathologists carried out an evaluation and, after further discussion of the evaluation, a staining decision was made. Additions have been made in this regard in the material and methods section.

L134. The staining strength was assessed by two pathologists, who then discussed their assessment and made a staining decision.

2) In section 3.1, It will be better to show the quantitative mIHC and sIHC results instead of only present the percentage.

>Thank you for the advice. We added total number of patients in the main text.

L184. sIHC analysis showed that 46 patients (44.7%) had high tumor B7-H3 expression, and 57 patients (55.3%) of 103 patients had low tumor B7-H3 expression. Furthermore, 48 (46.6%) and 55 (53.4%) patients of 103 patients had high and low stromal B7-H3 expressions, respectively.

3) In table 1, the subtitle and its content is not well designed. For example, the subtitle "P-vale" is located in the middle of two numbers.

>As the reviewer 1 suggested, we thought the design of table 1 was not very good. Table 1 and its title has been revised for clarity.

4) Authors show the correlation between B7-H3 and PD-L1 expression. Whether the data is from the whole group or its from high B7-H3 expression group?

> We apologize for the ambiguity of the subject. The data is from the whole group.

L.250 Finally, we examined the correlation between membrane B7-H3 and PD-L1 expres-sion in tumor and stromal cells by examining B7-H3 expression or PD-L1 expression lev-els per cell membrane pixel with slides from all groups.

Both B7-H3 and PD-L1 belongs to B7 family, it will not be very surprised to observe the positive correlation expression of B7-H3 and PD-L1. In addition, it looks like no significant difference between tumor, TMC, and stromal cells in Figure 4. Could authors claim the novelty or significance of this results?

>As the reviewer 1 said, there is room to analyze the expression of these molecules since they have similar functions. As far as we have been able to determine with BioGPS (http://biogps.org/), there is no strong correlation (correlation index greater than 0.5) between CD274 and 276. Although co-expression per cell has been confirmed in previous reports, this is the first study to show a quantitative correlation, and therein lies its novelty.

Reviewer 2 Report

Comments and Suggestions for Authors

REVIEWER`s COMMENTS

In the manuscript entitled “ Comprehensive characterization of B7-H3 expression in the tumour microenvironment of lung squamous cell carcinoma” by Asakawa and collaborators the Authors provide evidence that in lung squamous cell carcinomas (LSCCs) high expression in tumour cells of B7-H3/CD276, a regulator of T-cell-mediated immune responses (immune-checkpoint regulator), correlates with improved overall survival/OS in patients with stage II-III disease; high expression in stromal cells of the tumour microenvironment, however, provided no correlation with survival. A high Brinkman index (lifetime cigarette smoking load) was also found to correlate with higher expression of B7-H3. Additional findings were 1) a positive correlation between higher tumour cell expression of CD276 and increased densities of CD163+ macrophages (likely M2/proinflammatory-type) in the tumour stroma; 2) a positive correlation between lower stromal expression of CD276 and higher densities of intra-tumoral CD8+ T cells; and, 3) a strong correlation between plasma membrane expression of B7-H3/CD276 and PDL-1 (also an immune-checkpoint modulator), both in tumour and stromal (macrophage) cells.

The Authors used a high-end in situ technology (Multiplex immunohistochemistry) to determine protein expression and sub-cellular localization at the single cell level.

The main message in this piece of research, the finding of a positive correlation between high tumour cell expression of CD276 and better OS is potentially interesting but goes against mainstream data. Indeed, and as the Authors duly acknowledge, with the possible (and notable) exception of head and neck squamous cell carcinoma, overexpression of CD276/B7-H3 has been shown to correlate with poor prognosis in most cancers.

The comments of this Reviewer shall be thus viewed as an attempt to strengthen the data presented in this manuscript.

COMMENTS

A. General

A.1.  Type of study; this is a retrospective study, and this should be stated in the Abstract and throughout the main text; eventually, also in the Title.

A.2. The Abstract could be better structured to better highlight the main findings.

A.3. Although the Ms globally reads well, there are some parts that are confusing or even contradictory (cf further below, under specific remarks).

A.4. The Discussion section could be improved to better bridge some data that appear to contradict mainstream expectations (cf further below, under specific remarks).

A.5. At some point the Authors should clarify their choice for CD276/B7-H3 instead of other immune-checkpoint regulators.

B. SPECIFIC:

B.1. ANTIBOBY CHARACTERIZATION: this study relies heavily on the specific staining of B7-H3/CD276; however, it will be important to know to what extent the primary antibody recognizing CD276 has been characterized. We suggest that, in a limited set of samples (CD276-high and CD276-low), the Authors should compare staining of the antibody utilized herein with that of a reference antibody(ies). This is a relevant issue, to avoid the sad case of some publications on PD-1/PDL-1 that relied on poorly characterized antibodies, leading to much confusion and distrust in the field. Also, the animal species in which the antibodies used in this research were raised should be specified in Materials and Methods.

B.2. Line 186: the Brinkman index was referred without any prior explanation; to the reader who is naïve to the field providing an explanation will be helpful. Also, in the Discussion section this may be briefly discussed. Indeed, tobacco-associated cancers typically express copious amounts of neoantigens, rely more heavily on the host immune response, and may respond better to immunotherapies.

B.3. In Materials and Methods, the Authors should state how the estimation of cell densities (cells/mm2) was obtained; eg, how many cells counted per sample; also, if the cell densities (expressed in cells/ mm2) refer to the average of total area of tissue sections (tumour plus stromal cells; ie, cells per mm2 of tissue section), or else were adjusted to areas occupied by tumour cells vs areas comprising stroma (ie, cells per mm2 of tumour cells, cells per mm2 of stroma). This is important given that proportions between tumour cells and stromal cells are expectedly variable between samples.

B.4. Although not mentioned in the text of the Ms, the Reviewer believes that Authors are aware of the two isoforms described for CD276; although both isoforms (1 and 2) may negatively modulate CD4+ T cell responses, isoform 2 may enhance the cytotoxic function (and IFN-gamma production) of CD8+ T cells. This could be discussed in the Ms. Indeed, predominant expression of CD276 isoform 2 might explain the apparent paradox between higher expression of CD276/B7-H3 and better clinical outcome found herein for lung squamous cell carcinomas (LSCCs).

B.5. CONTRADICTORY MESSAGES: In RESULTS, the text in Lines 224-231 (““Relationship between stromal B7-H3 expression and cell phenotypes:  Next, we determined whether stromal B7-H3 expression also affected different cell  phenotypes by measuring the cell densities of TILs (CD4+PD-1-, CD4+PD-1+, CD8+PD-1-, and CD8+PD-1+ cells) and TAMs (CD68+PD-L1-, CD68+PD-L1+, CD163+PD-L1-, and CD163+PD-L1+ cells). The counts of CD8+PD-1- cells were significantly higher in the tumour area in the high B7-H3 expression group than that in the low B7-H3 expression group (P 229 = 0.0262); however, no other phenotypes were significantly affected in either area (Supplementary Figure 1”” sic) IS CONTRADICTORY to the text in DISCUSSION, between Lines 285-288 (“”When we stratified tumor and stromal TILs and TAMs by B7-H3 expression, we  found no significant difference in intratumoral CD8+ T-cell infiltration in the group with low tumor B7-H3 expression, whereas the group with low stromal B7-H3 expression had  a significantly higher density of intratumoral CD8+ T cells”” sic).

In RESULTS a higher expression of B7-H3 in stroma correlated with higher tumour infiltration by CD8+ T cells, in DISCUSSION, it was a lower expression of B7-H3 in the stroma that correlated with the CD8+ T cell infiltration!!

 Looking at the data in Supplemental Figure 1 (k), it seems that the correlation is indeed between lower stromal expression (of B7-H3) and higher CD8+ T cell infiltration of the tumour cells (as assumed in Discussion).

This is the type of writing gaffe that should not occur.

B.6. In Figures 4 and 5 (main text), the graphs could benefit in clarity by adding a small text, eg,  Tumor cells in Fig. 4 (a), Stromal cells in Fig. 4 (c) embedded in the figure. This would avoid resorting exclusively to the figure legend.

B.7. In line 207 of main text (“”To determine whether tumor B7-H3 expression affected the pattern of cell phenotypes, we compared the cell density of each cell phenotype in samples from patients with high and low tumor B7-H3 expression”” sic). This study is observational, not experimental. Therefore, it is methodologically incorrect to attempt at establishing cause-effect relationships between B7-H3 expression and cell phenotypes. Instead of “affected the pattern” it would be wiser to write something like “correlated with the pattern” or “was associated with a pattern”.

B.8. Lines 330-332: “”In patients with NSCLC, the current outcome of PD-L1 treatment is 84.3% when treat-ment can be completed, but the outcome is only 27.3–39.1% in cases where treatment can not be completed””. What do Authors mean by current outcome; favourable outcome??

B.9. Lines 348-350: “”Furthermore, several markers were examined in the quantitative  search for stromal cells; however, it was not possible to stain some markers for stromal cells that may also have been affected by B7-H3 expression””. What do Authors exactly mean?

Comments on the Quality of English Language

Some sentences might be improved for the sake of clarity. Some of these pieces of text are highlighted in our comments to the Authors.

Author Response

Reviewer 2

In the manuscript entitled “ Comprehensive characterization of B7-H3 expression in the tumour microenvironment of lung squamous cell carcinoma” by Asakawa and collaborators the Authors provide evidence that in lung squamous cell carcinomas (LSCCs) high expression in tumour cells of B7-H3/CD276, a regulator of T-cell-mediated immune responses (immune-checkpoint regulator), correlates with improved overall survival/OS in patients with stage II-III disease; high expression in stromal cells of the tumour microenvironment, however, provided no correlation with survival. A high Brinkman index (lifetime cigarette smoking load) was also found to correlate with higher expression of B7-H3. Additional findings were 1) a positive correlation between higher tumour cell expression of CD276 and increased densities of CD163+ macrophages (likely M2/proinflammatory-type) in the tumour stroma; 2) a positive correlation between lower stromal expression of CD276 and higher densities of intra-tumoral CD8+ T cells; and, 3) a strong correlation between plasma membrane expression of B7-H3/CD276 and PDL-1 (also an immune-checkpoint modulator), both in tumour and stromal (macrophage) cells.

The Authors used a high-end in situ technology (Multiplex immunohistochemistry) to determine protein expression and sub-cellular localization at the single cell level.

The main message in this piece of research, the finding of a positive correlation between high tumour cell expression of CD276 and better OS is potentially interesting but goes against mainstream data. Indeed, and as the Authors duly acknowledge, with the possible (and notable) exception of head and neck squamous cell carcinoma, overexpression of CD276/B7-H3 has been shown to correlate with poor prognosis in most cancers.

The comments of this Reviewer shall be thus viewed as an attempt to strengthen the data presented in this manuscript.

COMMENTS

  1. General

A.1.  Type of study; this is a retrospective study, and this should be stated in the Abstract and throughout the main text; eventually, also in the Title.

>Thank you for pointing that out. We added “: A retrospective study” in the Title and the Abstract. In the main text, we mentioned that in chapter 2.1.

L.98 The clinicopathological characteristics of the cohort were collected retrospectively based on patient medical records and surgical pathology reports.

A.2. The Abstract could be better structured to better highlight the main findings.

> Thank you for the advice. The main point of this study is to show a quantitative correlation between B7-H3 and PD-L1 in the same cell, so we rewrote it to emphasize this point.

Abstract. Quantitative analysis revealed that there is a positive correlation between B7-H3 and PD-L1 ex-pression in tumor and stromal cells, as well as in intratumoral tumor-infiltrating lymphocytes and tumor-associated macrophages in the same cells. CD68+, CD163+, and CK+ cells with PD-L1+ phenotypes had higher B7-H3 expression compared to PD-L1- cells.

A.3. Although the Ms globally reads well, there are some parts that are confusing or even contradictory (cf further below, under specific remarks).

>As the reviewer 2 pointed out, there were some inconsistencies in the text. This has been corrected. Details are listed below (B SPECIFIC).

A.4. The Discussion section could be improved to better bridge some data that appear to contradict mainstream expectations (cf further below, under specific remarks).

>Thank you for pointing that out. The reviewer 2 indicated, we have corrected the inconsistencies in the Discussion section. Details are listed below (B SPECIFIC).

A.5. At some point the Authors should clarify their choice for CD276/B7-H3 instead of other immune-checkpoint regulators.

>As the reviewer 2 said, there are many other factors that are candidates for Immune checkpoint inhibitor (ICI) targets, such as CTLA4. PD-L1 and B7-H3 are similar molecules in terms of ICI, but inhibition of PD-L1 access alone is not currently sufficient for therapeutic efficacy. From this perspective, we focused on B7-H3 in this study as a factor that complements or is related to PD-L1 inhibition.

  1. SPECIFIC:

B.1. ANTIBOBY CHARACTERIZATION: this study relies heavily on the specific staining of B7-H3/CD276; however, it will be important to know to what extent the primary antibody recognizing CD276 has been characterized. We suggest that, in a limited set of samples (CD276-high and CD276-low), the Authors should compare staining of the antibody utilized herein with that of a reference antibody(ies). This is a relevant issue, to avoid the sad case of some publications on PD-1/PDL-1 that relied on poorly characterized antibodies, leading to much confusion and distrust in the field. Also, the animal species in which the antibodies used in this research were raised should be specified in Materials and Methods.

>Thank you for the valuable comments. This B7-H3 antibody is a mouse monoclonal antibody produced by Translational Medicine & Clinical Pharmacology Department, Daiichi Sankyo Co. Ltd.. Antibodies developed as companion reagents for therapeutic strategies and have been previously reported (Lung Cancer. 2017:103:44-51). On that premise, Daiichi Sankyo Co. Ltd. has actually conducted sufficient validation and has data, but unfortunately, we are not able to make it public because we are considering filing a patent application for the antibody. The point that this antibody is a mouse monoclonal antibody has been added in the text.

L.117 antibodies against B7-H3 (mouse monoclonal antibody, 1:600, BD5A11, Daiichi Sankyo)

B.2. Line 186: the Brinkman index was referred without any prior explanation; to the reader who is naïve to the field providing an explanation will be helpful. Also, in the Discussion section this may be briefly discussed. Indeed, tobacco-associated cancers typically express copious amounts of neoantigens, rely more heavily on the host immune response, and may respond better to immunotherapies.

>Thank you for the useful advices. The Brinkman index is a numerical value used to quantify an individual’s cumulative exposure to tobacco smoke. It is calculated by multiplying the number of cigarettes smoke per day by the number of years the person has been smoking. We added the explanation of the Brinkman Index in the main text. Then, we added a note about the link between tobacco and immunity in the discussion part.

L.188 The Brinkman index, which is calculated by multiplying the number of cigarettes smoke per day by the number of years the person has been smoking, was significantly higher in the high tumor B7-H3 expression (P = 0.0258) and high stromal B7-H3 expression (P = 0.0142) groups than that in the low-expression group.

L.291 The present cohort includes many smokers. Smoking-related cancers may express larger amounts of neoantigen.[19] If the analysis were performed in a cohort of patients with ICI, different results could be obtained.

B.3. In Materials and Methods, the Authors should state how the estimation of cell densities (cells/mm2) was obtained; eg, how many cells counted per sample; also, if the cell densities (expressed in cells/ mm2) refer to the average of total area of tissue sections (tumour plus stromal cells; ie, cells per mm2 of tissue section), or else were adjusted to areas occupied by tumour cells vs areas comprising stroma (ie, cells per mm2 of tumour cells, cells per mm2 of stroma). This is important given that proportions between tumour cells and stromal cells are expectedly variable between samples.

>We totally agree with reviewer 2. It is the average of total area of tissue sections. Adjusting for tumor to stromal ratio may yield more reliable data. We added this in the limitations.

L 375. The proportion of tumor and stromal regions in ROI is not uniform. Rigorous adjustment of the proportions could provide more reliable data.

B.4. Although not mentioned in the text of the Ms, the Reviewer believes that Authors are aware of the two isoforms described for CD276; although both isoforms (1 and 2) may negatively modulate CD4+ T cell responses, isoform 2 may enhance the cytotoxic function (and IFN-gamma production) of CD8+ T cells. This could be discussed in the Ms. Indeed, predominant expression of CD276 isoform 2 might explain the apparent paradox between higher expression of CD276/B7-H3 and better clinical outcome found herein for lung squamous cell carcinomas (LSCCs).

>It is possible to recognize two isoforms, but unfortunately, we have not been able to confirm which of the two isoforms the antibody we used recognizes. As the reviewer 2 mentioned, if this antibody were to recognize isoform 2, it would be expected to attack tumor cells by enhancing CD8-positive T cells, resulting in a better prognosis for the patient, which is consistent with the results of this study. We added this point in discussion and a reference.

  1. 298 Furthermore, there are two isoforms of B7-H3 [22], but it has not been confirmed which isoform the B7-H3 used in this study recognizes. If this antibody were to recognize isoform 2, it would be expected to attack tumor cells by enhancing CD8-positive T cells, resulting in a better prognosis for the patient, which is consistent with the results of this study. This may be one of the reasons why the results of this study differed from those previously reported.

B.5. CONTRADICTORY MESSAGES: In RESULTS, the text in Lines 224-231 (““Relationship between stromal B7-H3 expression and cell phenotypes:  Next, we determined whether stromal B7-H3 expression also affected different cell  phenotypes by measuring the cell densities of TILs (CD4+PD-1-, CD4+PD-1+, CD8+PD-1-, and CD8+PD-1+ cells) and TAMs (CD68+PD-L1-, CD68+PD-L1+, CD163+PD-L1-, and CD163+PD-L1+ cells). The counts of CD8+PD-1- cells were significantly higher in the tumour area in the high B7-H3 expression group than that in the low B7-H3 expression group (P 229 = 0.0262); however, no other phenotypes were significantly affected in either area (Supplementary Figure 1”” sic) IS CONTRADICTORY to the text in DISCUSSION, between Lines 285-288 (“”When we stratified tumor and stromal TILs and TAMs by B7-H3 expression, we  found no significant difference in intratumoral CD8+ T-cell infiltration in the group with low tumor B7-H3 expression, whereas the group with low stromal B7-H3 expression had  a significantly higher density of intratumoral CD8+ T cells”” sic).

In RESULTS a higher expression of B7-H3 in stroma correlated with higher tumour infiltration by CD8+ T cells, in DISCUSSION, it was a lower expression of B7-H3 in the stroma that correlated with the CD8+ T cell infiltration!!

 Looking at the data in Supplemental Figure 1 (k), it seems that the correlation is indeed between lower stromal expression (of B7-H3) and higher CD8+ T cell infiltration of the tumour cells (as assumed in Discussion).

This is the type of writing gaffe that should not occur.

>Thank you for pointing out our carelessness. As the reviewer 2 indicated, the RESULT was incorrect and we have rewritten it as follows.

L.245 The counts of CD8+PD-1- cells were significantly higher in the tumor area in the low B7-H3 expression group than that in the high B7-H3 expression group (P = 0.0262); however, no other phenotypes were significantly affected in either area (Supplementary Figure 1).

B.6. In Figures 4 and 5 (main text), the graphs could benefit in clarity by adding a small text, eg,  Tumor cells in Fig. 4 (a), Stromal cells in Fig. 4 (c) embedded in the figure. This would avoid resorting exclusively to the figure legend.

>Thank you for pointing that out. We added text in the figure 4 and 5.

B.7. In line 207 of main text (“”To determine whether tumor B7-H3 expression affected the pattern of cell phenotypes, we compared the cell density of each cell phenotype in samples from patients with high and low tumor B7-H3 expression”” sic). This study is observational, not experimental. Therefore, it is methodologically incorrect to attempt at establishing cause-effect relationships between B7-H3 expression and cell phenotypes. Instead of “affected the pattern” it would be wiser to write something like “correlated with the pattern” or “was associated with a pattern”.

>Thank you for the advice. The word "affected" has been changed to "associated" because this is a retrospective study, as the reviewer 2 mentioned.

B.8. Lines 330-332: “”In patients with NSCLC, the current outcome of PD-L1 treatment is 84.3% when treat-ment can be completed, but the outcome is only 27.3–39.1% in cases where treatment can not be completed””. What do Authors mean by current outcome; favourable outcome??

>As the reviewer 2 mentioned, it was unclear. This indicates a good prognosis if treatment can be completed, but a poor prognosis if treatment cannot be completed. We have specifically described the outcome.

L358. This indicates that the prognosis is good if treatment is completed, while the prognosis is poor if treatment is not completed

B.9. Lines 348-350: “”Furthermore, several markers were examined in the quantitative  search for stromal cells; however, it was not possible to stain some markers for stromal cells that may also have been affected by B7-H3 expression””. What do Authors exactly mean?

> Thank you for the comments. Stromal cells are composed of pathologically diverse cells such as vascular endothelial cells, fibroblasts, and blood cells. Since we could not perform immunostaining for each marker, we could not examine the B7-H3 expression of the constituent cells in detail. Therefore, we have added this point to the limitation.

L.377. however, it was not possible to stain some markers for stromal cells, including vascular endothelial cells, fibroblasts, and blood cells, that may also have been affected by B7-H3 expression.

Reviewer 3 Report

Comments and Suggestions for Authors

1. In Figure 2, the survival curves are unreadable; the quality of the figures needs to be improved. Why are survival curves shown in the Methods section and not in the Results section?

2. Make Table 1 a table rather than a figure.

3. I didn’t see any differences in Table 1, maybe it would make sense to divide it into 3 subgroups: low, medium and high expression, adding an intermediate subgroup?

4. I would like to see the survival analysis in more detail, including univariate analysis (add a table) + multifactorial (transfer from additional materials to the main part). It seems to me that sections 3.2 and 3.3 should be more detailed, since the maximum differences are found there.

5. Figure 3 shows no differences between subgroups with 1 exception. Does it make sense to show all the pictures? What information do they carry?

6. Why is the expression of PD-L1, PD-1, CD4, etc. not taken into account in the survival analysis? Maybe the impact on survival rates will be more pronounced?

Author Response

Reviewer 3

  1. In Figure 2, the survival curves are unreadable; the quality of the figures needs to be improved. Why are survival curves shown in the Methods section and not in the Results section?

>We agree with the reviewer 3’s comment. We changed the location of Figure 2 in Results section. 

  1. Make Table 1 a table rather than a figure.

> Thank you for pointing that out. Table 1 and the title of Table 1 has been revised for clarity.

  1. I didn’t see any differences in Table 1, maybe it would make sense to divide it into 3 subgroups: low, medium and high expression, adding an intermediate subgroup?

>Thank you for the suggestion. The cohort of approximately 100 cases of lung squamous cell carcinoma has been designed with the hypothesis that classification into two groups would lead to statistical differences, whereas classification into three groups would allow a more detailed study. However, it was difficult to formulate a hypothesis, and the classification and evaluation was conducted in two groups.

  1. I would like to see the survival analysis in more detail, including univariate analysis (add a table) + multifactorial (transfer from additional materials to the main part). It seems to me that sections 3.2 and 3.3 should be more detailed, since the maximum differences are found there.

>As the reviewer 3 mentioned, we have significant data, so we changed univariate and multivariate results table to the main table, and added some explanations in the section 3.2 and 3.3.

  1. Figure 3 shows no differences between subgroups with 1 exception. Does it make sense to show all the pictures? What information do they carry?

> Thanks for the suggestion, as the reviewer 3 mentioned, no statistically significant difference is observed in Fig. 3 except for CD163+PD-L1(stroma). The number of positive cells for various markers, including the duplex ones considered in this study, were each analyzed as an immunologically significant subset. As a result, we consider the lack of significant differences to be of a certain significance and have included all data.

  1. Why is the expression of PD-L1, PD-1, CD4, etc. not taken into account in the survival analysis? Maybe the impact on survival rates will be more pronounced?

>Thank you for the suggestion. As for PD-L1 and PD-1 and CD4, these have also been reported as important factors. (ref: PD-1: Turk Gogus Kalp Damar Cerrahisi Derg. 2024 Jan 29;32(1):84-92., PD-L1: Cancer Manag Res. 2021 Aug 12:13:6365-6375., PD-L1: J Thorac Oncol. 2019 Jan;14(1):25-36., CD4: Genome Med. 2022 Jul 8;14(1):72.) In this study, we focused on B7-H3, which has not yet been reported.

Round 2

Reviewer 3 Report

Comments and Suggestions for Authors

1. The authors never corrected the quality of the survival curves in Figure 2, only changing it to another place in the text of the manuscript.

2. All figures have an extra inscription with the figure number and the authors of the manuscript. For what?

3. Tables are still presented in the form of pictures and the table title should be at the top.

4. Figures 5b, c are very small, the quality needs to be improved.